# Safety of íntravitreal panitumumab combined with intravitreal ranibizumab in rabbits

Mukharram M. Bikbov[1], Gyulli M. Kazakbaeva[1], Iskander D. Valishin[1],
Aigul M. Ishbulatova[1], Songhomitra Panda-Jonas[2,3,4,5], Gulshat R. Mustafina[1],
Jost B. Jonas [3,4,5,6,7,8,9]*

1 Ufa Eye Research Institute of Bashkir State Medical University, Ufa, Russia, 2 Department of Ophthalmology, University Hospital Heidelberg, Heidelberg, Germany, 3 Rothschild Foundation Hospital, Institut Français de Myopie, Paris, France, 4 LV Prasad Eye Institute, Hyderabad, Telangana, India, 5 Privatpraxis Prof Jonas und Dr Panda-Jonas, Heidelberg, Germany, 6 Singapore Eye Research Institute, Singapore National Eye Center, Singapore, Singapore, 7 Beijing Visual Science and Translational Eye Research Institute (BERI), Beijing Tsinghua Changgung Hospital, Tsinghua Medicine, Tsinghua University, Beijing, China, 8 Department of Ophthalmology and Visual Sciences, The Chinese University of Hong Kong, Hong Kong, 9 New York Eye and Ear Infirmary of Mount Sinai, Icahn School of Medicine at Mount Sinai, New York, New York, United States of America

* Jost.Jonas@medma.uni-heidelberg.de

## Abstract

### Purpose

To examine the safety of intravitreally applied epidermal growth factor (EGF) receptor antibody panitumumab combined with intravitreally injected vascular endothelial growth factor (VEGF) antibody ranibizumab in rabbits.

### Methods

The experimental study included 8 male rabbits (age: 2−3 months; weight: 2.5–2.6 kg). The right eyes received three intravitreal injections of panitumumab (1 mg in 0.10 mL) combined with intravitreal injections of ranibizumab (0.5 mg in 0.05 mL) in intervals of 4 weeks, while the left eyes received three double intravitreal injections of Ringer´s solution (0.10 mL and 0.05mL). At baseline, at the time points of the re-injections, and at 2 weeks and at 6 weeks after the last injection, the animals were examined by fundus photography and optical coherence tomography (OCT). At study end at 2 weeks after the last injection for 4 animals (total study duration: 10 weeks) or at 6 weeks after the last injection for the remaining 4 rabbits (total study duration: 14 weeks), the animals were sacrificed and the eyes were histomorphometrically examined.

### Results

The right eyes (study eyes) and left eyes (control eyes) did not differ significantly in cell count of the retinal ganglion cell layer (8.5 ± 8.3 cells per 150μm

**Data availability statement:** The datasets with the microdata has been uploaded together with the paper and is thus publicly available.

**Funding:** The author(s) received no specific funding for this work.

**Competing interests:** It is also declared: Conflict of interest disclosure: Jost B. Jonas, Songhomitra Panda-Jonas: European patent EP 3 271 392, JP 2021-119187, and US 2021 0340237 A1: "Agents for use in the therapeutic or prophylactic treatment of myopia or hyperopia; European patent application 23196899.1 „EGFR Antagonists for the treatment of diseases involving unwanted migration, proliferation, and metaplasia of retinal pigment epithelium (RPE) cells". This does not alter our adherence to PLOS ONE policies on sharing data and materials. All other authors: None.

versus $4.5 \pm 3.3$ cells; $P = 0.04$), inner nuclear layer ($33.1 \pm 14.0$ cells versus $49.0 \pm 22.3$ cells; $P = 0.04$) and outer nuclear layer ($133.7 \pm 50.0$ cells versus $171.3 \pm 75.3$ cells; $P = 0.07$) nor in mean thickness of the ganglion cell layer ($31.8 \pm 34.4$ μm versus $24.3 \pm 15.2$ μm; $P = 0.15$), inner plexiform layer ($32.4 \pm 48.2$ μm versus $25.6 \pm 11.2$ μm; $P = 0.22$), inner nuclear layer ($26.6 \pm 14.0$ μm versus $26.7 \pm 10.2$ μm; $P = 0.43$), outer plexiform layer ($6.6 \pm 3.7$ μm versus $6.8 \pm 2.0$ μm; $P = 0.07$), outer nuclear layer ($32.6 \pm 16.1$ μm versus $30.6 \pm 11.8$ μm; $P = 0.82$) and of total retina ($157.7 \pm 99.4$ μm versus $152.0 \pm 60.7$ μm; $P = 0.34$). Apoptotic cells were not detected in either group. Mean IOP did not differ significantly between the right eyes (study eyes) and the left eyes (control eyes). Intravital signs of intraocular toxicity were not detected.

## Conclusions

Repeatedly intraocularly applied panitumumab combined with ranibizumab was intraocularly well tolerated and did not differ in the ocular reaction from intravitreal injections of Ringer´s solution.

---

### Introduction

Exudative maculopathies such as neovascular age-related macular degeneration (AMD), polypoidal choroidal vasculopathy and neovascular myopic macular degeneration are characterized by the formation of new vessels growing from the choroid into the space beneath the retinal pigment epithelium (RPE), or after breaking through the RPE reaching into the subretinal space in the macular region [1–3]. After the clinical introduction of intravitreal injections, intravitreally applied anti-vascular endothelial growth factor (VEGF) antibodies (such as ranibizumab, bevacizumab, aflibercept, brolucizumab, and faricimab) have profoundly changed the treatment of neovascular maculopathies and have substantially improved the therapeutical success [4–7]. By the induced regression of the newly formed blood vessels, the intravitreal anti-VEGF therapy resulted in a temporary or long-lasting reduction of sub-RPE, subretinal and intraretinal edema and corresponding improvements in central visual acuity, as compared to untreated controls [5–7].

The neovascularization beneath and on top of the RPE is often accompanied by a proliferation of RPE cells which undergo an epithelial-mesenchymal transition and form a subretinal fibrous layer separating the retinal photoreceptors from the nourishing choriocapillaris. In addition, the transformed RPE cells, although still in contact with a basal membrane and thus preserving some epithelial characteristic, can no longer fully serve the photoreceptors by providing functions such as phagocytosis of the photoreceptor outer segments and regeneration of photopigments [8,9]. The formation of the subretinal fibrous scar is irreversible and has, besides an indirect effect though an anti-neovascular treatment, therapeutically not directly been addressed so far. In cell culture studies, RPE cells have extensively been examined with respect to their physiology and pathophysiology including molecules influencing through cell

surface receptors the RPE cell migration and proliferation [10–18]. These investigations revealed that epidermal growth factor (EGF) is a strong promotor of the proliferation and migration of RPE cells, while blocking the EGF receptor reduces the RPE cell proliferation and migration [10–18]. Also based on the clinical finding that eyes with exudative age-related macular degeneration and myopic macular neovascularization have an increased intraocular concentration of EGF, it led to the hypothesis, that adding EGF receptor (EGFR) antibodies to VEGF antibodies in the therapy of neovascular proliferative maculopathies may be useful to reduce the unwanted subretinal RPE cell proliferation [19,20]. Here we examined the intraocular safety of a combined intravitreal application of an anti-VEGF drug and an anti-EGFR molecule as a preparatory step towards a potential clinical application.

## Materials and methods

The experimental study included rabbits (male gray Soviet chinchilla; age: 2−3 months; weight: 2.0–2.6 kg), which were treated in accordance with the ARVO (Association for Research in Vision and Ophthalmology) Statement for the Use of Animals in Ophthalmic and Vision Research. The investigation was approved by the Ufa Eye Research Institute Biomedical Ethics Committee and confirmed that the study was conducted in accordance with the ARRIVE (Animal Research: Reporting of In Vivo Experiments) guidelines. As already described in detail previously, the animals were obtained from a commercial vendor (Federal State Unitary Enterprise "Scientific and Production Association for Immunological Preparations "Microgen" of the Ministry of Health of the Russian Federation, Ufa, Bashkortostan, Russia) [21]. They were kept at a constant room temperature (22 ± 1°C) and in a light-controlled environment (lights on from 7 am to 7 pm) with ad libitum access to food and water.

The right eyes of the animals received an intravitreal injection of 1 mg of panitumumab (Vectibix®) in 0.10 mL combined with a second intravitreal injection of 0.5 mg ranibizumab (Lucentis®) in 0.05 mL. The contralateral left eyes received a double intravitreal injection of 0.10 mL Ringer´s solution and of 0.05 ml of Ringer's solution (i.e., two separate injections) (Gematek OOO Company, Moscow, Russia). The injections were repeated twice in an interval of 4 weeks. Directly after each injection, we measured the intraocular pressure (IOP) of both eyes. The injections were performed under general anesthesia which was achieved by an intramuscular injection (biceps femoris) of Zoletil (15 mg/kg) (tiletamine mixed with zolazepam; Valdepharm Co., Val-de-Reuil, France) and xylazine (20 mg/kg) (Xyla; Interchemie Werken, De Adelaar B.V., A Waalre, The Netherlands). We additionally applied anesthetic eye drops (0.4% oxybuprocain, Inocain; Promed Exports, New Delhi, India) topically. The injections were carried out in the temporal upper quadrant of the eyes at a distance of 3–4 mm from the limbus. The technique has been described recently in detail [21].

At baseline, at the time of the re-injections, and at 14 days after the third (last) injection, all animals were examined. The examinations included an inspection and photography of the external eye, fundus photography (VISUCAM 500, Carl Zeiss Meditec AG, Jena Germany), optical coherence tomography (OCT) of the fundus, and tonometry (Auto-2Ref/Keratometer HRK-7000A Huvitz Co, Ltd., Gyeonggi-do, Korea). Every second rabbit was sacrificed at two weeks after the last injection (total study duration: 10 weeks), while the remaining animals were re-examined 4 weeks later and sacrificed (total study duration: 14 weeks). At the time of sacrificing the animals, the eyes were enucleated under deep general anesthesia, and the rabbits were sacrificed by injecting air (5–10 mL) into the ear vein.

As also described in detail previously, the enucleated eyes were histologically processed for light microscopy (Fig 1a-1d) [21]. We determined the horizontal, vertical and sagittal diameters of the fixed eyes [21]. A central part of the globes with a thickness of approximately 8 mm was removed. It included the pupil and the optic nerve head. After dehydration in alcohol, the tissue was imbedded in paraffin, and we prepared histological slides with a thickness of 4 μm to 6 μm. The slides were stained with hematoxylin eosin. Using a light microscope, we counted the cell numbers in the retinal ganglion cell layer, the inner nuclear layer, and the outer retinal nuclear layer, and we measured the thickness of the ganglion cell layer, inner plexiform layer, inner nuclear layer, outer plexiform layer, outer nuclear layer and of the total retina. We additionally stained the slides for apoptotic cell death using the TUNEL (terminal deoxynucleotidyl transferase dUTP nick end labeling) technique (Figs 2, 3) [21].

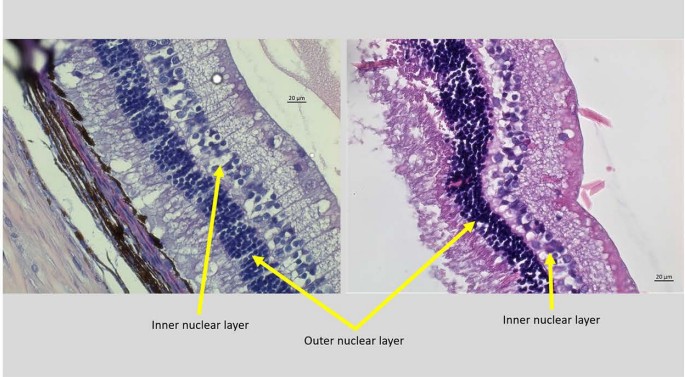
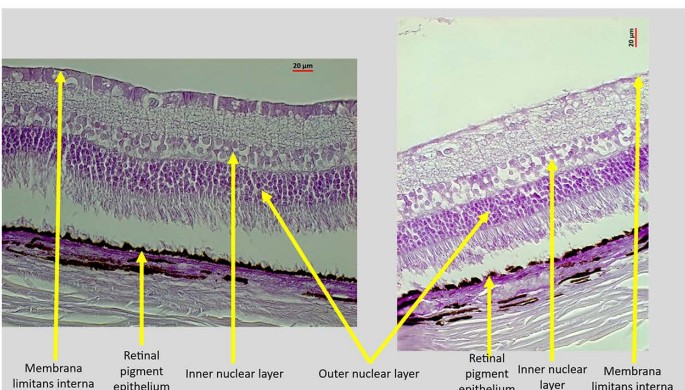
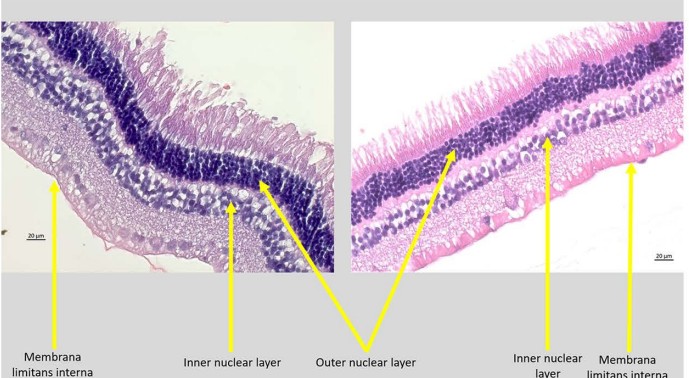
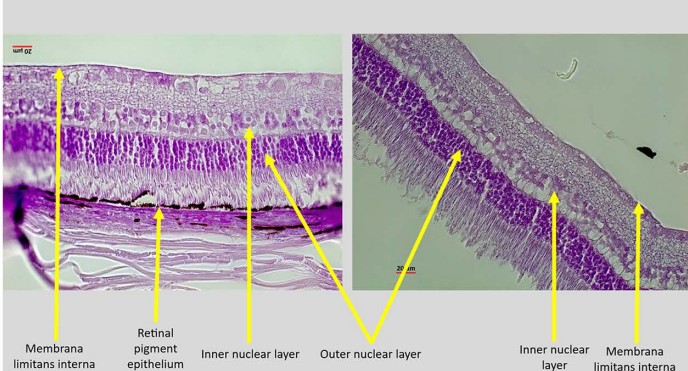

**Fig 1. Histo-photographs of the retina of right eyes (study group; on the left side of the image) of rabbits after repeated intravitreal combined injections of panitumumab (1 mg in 0.10 mL) ranibizumab (0.5 mg in 0.05 mL) in intervals of 4 weeks, and of the corresponding left eyes (control eyes; on the right side of the image) receiving a double intravitreal injection of 0.10 mL Ringer´s solution and of 0.05 ml of Ringer's solution (i.e., two separate injections).** The slides were stained by hematoxylin-eosin. a,b,c,d.

In the statistical analysis (SPSS for Windows, version 27.0, IBM-SPSS, Chicago, IL, USA), we assessed the mean values ± their standard deviations of the main outcome parameters, i.e., IOP, and histomorphometric parameters. Using the Wilcoxon test for paired samples, we compared the measurements obtained in the right eyes versus those of the left eyes. An analysis of variance was additionally performed. A two-sided *P*-value was considered statistically significant if it was smaller than 0.05.

## Results

The study included 8 rabbits which were divided in the group with 10 weeks study duration (n = 4) and the group with a study duration of 14 weeks (n = 4).

Including the total study population into the analysis, the right eyes (study eyes) and left eyes (control eyes) did not differ significantly in cell count of the retinal ganglion cell layer, inner nuclear layer and outer retinal nuclear layer nor in the thickness of the ganglion cell layer, inner plexiform layer, inner nuclear layer, outer plexiform layer, outer nuclear layer and of total retina (Table 1). The same result was obtained if the study population was stratified into the subgroup with a total study duration of 10 weeks and a subgroup with a total study duration of 14 weeks (Fig 1a-1d). In an analysis of variance, similar findings were made (Table 1). There was a borderline statistical significance between the study group (right eyes) and control group (left eyes), showing a slightly higher ganglion cell layer cell count in the study group as compared to the

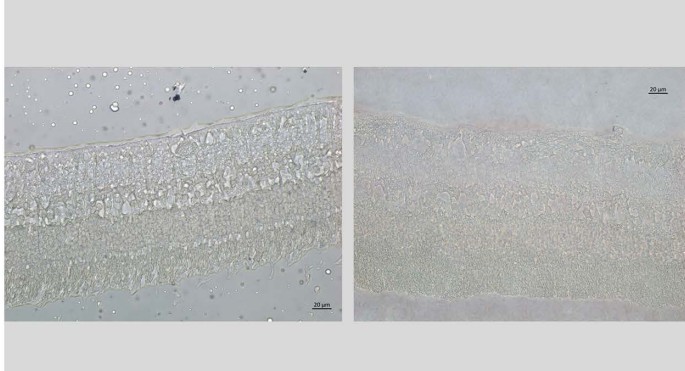
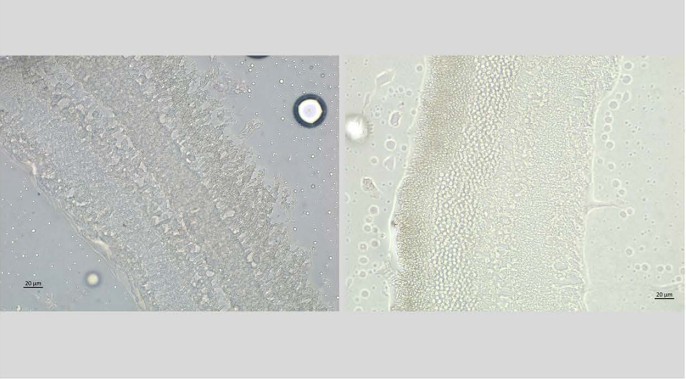
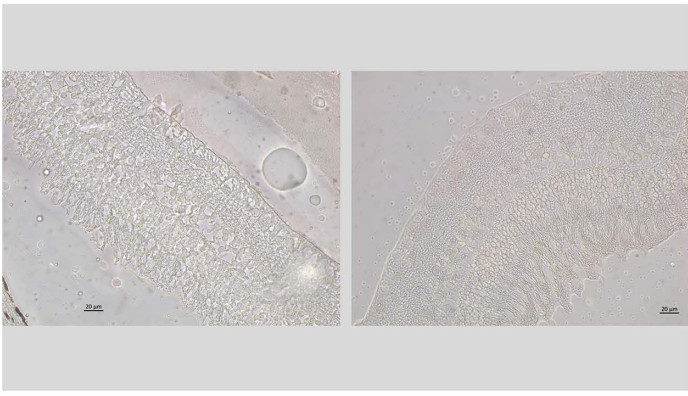

**Fig 2. Histo-photograph showing images of the retina of the right eyes (study group; left side of the image) of different rabbits after repeated intravitreal combined injections of panitumumab (1 mg in 0.10 mL) ranibizumab (0.5 mg in 0.05 mL) in intervals of 4 weeks, and of the corresponding left eyes (control eyes; right side of the image) receiving a double intravitreal injection of 0.10 mL Ringer´s solution and of 0.05 ml of Ringer's solution (i.e., two separate injections).** The slides were stained by the TUNEL (terminal deoxynucleotidyl transferase dUTP nick end labeling) technique. None of the slides showed a TUNEL-positive cell. a: Images from the equatorial region. b: Images from the equatorial region. c: Images from the posterior pole.

control group (8.5±8.3 versus 4.5±3.3; $P=0.04$ (Wilcoxon test) and $P=0.06$ (ANOVA)), while in the inner nuclear layer cell count was lower in the study group than in the control group (33.1±14.0 versus 49.0±22.3; $P=0.04$ (Wilcoxon test) and $P=0.02$ (ANOVA)) (Table 1). After conducting a Bonferroni´s correction for performing multiple statistical analyses, all differences between the study group and control group were not statistically significant.

The TUNEL-stained slides did not show apoptotic cells neither in the eyes of the study group nor in those of the control group (Figs 2, 3).

Mean IOP did not differ significantly between the right eyes (study eyes) and the left eyes (control eyes) (Table 2). The injection-related IOP rise after the first injections was 25.0±16.2 mm Hg in the right eyes and 20.7±19.0 mm Hg in the left eyes, without a significant inter-eye difference ($P=0.41$) (Table 2). Similar results were obtained for the IOP readings after the second and third injections (Table 2).

Two rabbits of the short duration group developed bilateral cataracts so that the fundus could no longer be visualized at the end of the study period. Besides lens opacities, the fundus photographs and OCT images did not reveal any signs of intraretinal inflammation (Fig 4). Biomicroscopical examinations of the anterior chamber neither revealed any signs of intraocular inflammation.

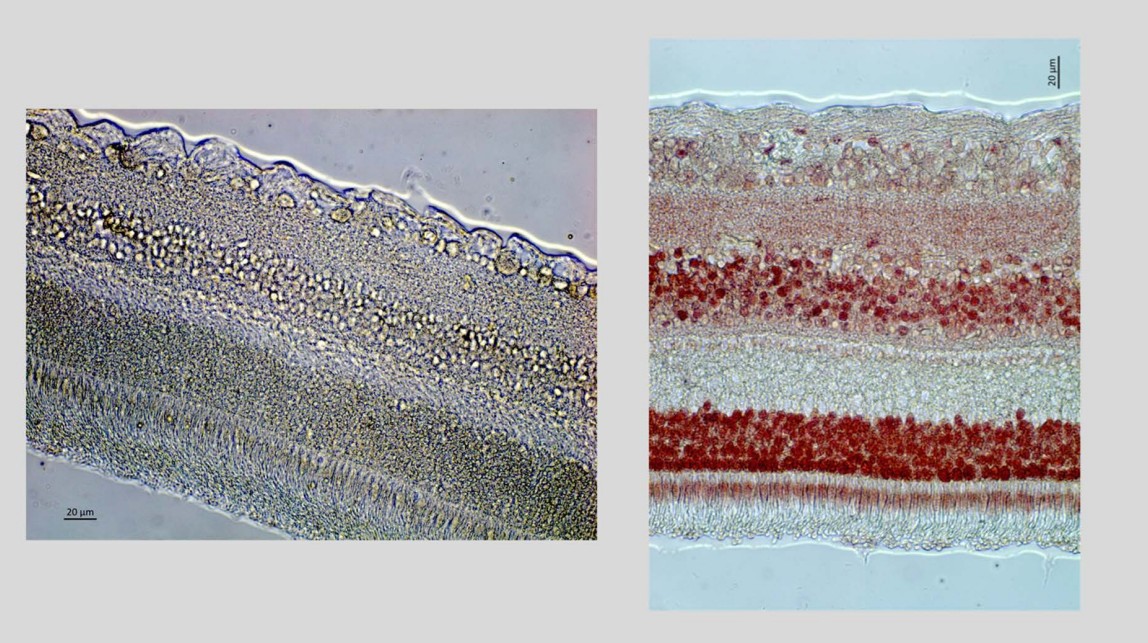

**Fig 3. Histo-photograph showing images of the retina of human eyes of a negative control (left side of the image) and a positive control (right side of the image) using the TUNEL (terminal deoxynucleotidyl transferase dUTP nick end labeling) technique for staining.**

**Table 1. Retinal histomorphometric measurements (mean±standard deviation) in rabbits receiving intravitreal injections of panitumumab combined with ranibizumab (right eyes) and of Ringer´s solution (left eyes). The differences between both eyes were statistically not significant (with applying Bonferroni´s correction for performing multiple statistical analyses).**

| Parameter | Right eyes (study group) | Left eyes (control group) | *P*-value (Wilcoxon-test) | *P*-value (analysis of variance) |
|---|---|---|---|---|
| Ganglion cell layer cell count | 8.5±8.3 | 4.5±3.3 | 0.04 | 0.06 |
| Ganglion cell layer thickness (µm) | 31.8±34.5 | 24.3±15.2 | 0.15 | 0.15 |
| Inner plexiform layer thickness (µm) | 32.4±48.2 | 25.6±11.2 | 0.22 | 0.32 |
| Inner nuclear layer cell count | 33.1±14.0 | 49.0±22.3 | 0.04 | 0.02 |
| Inner nuclear layer thickness (µm) | 26.6±14.0 | 26.7±10.2 | 0.43 | 0.98 |
| Outer plexiform layer thickness (µm) | 6.6±3.7 | 6.8±2.0 | 0.07 | 0.73 |
| Outer nuclear layer cell count | 133.7±50.0 | 171.3±75.3 | 0.07 | 0.11 |
| Outer nuclear layer thickness (µm) | 32.6±16.1 | 30.6±11.8 | 0.82 | 0.49 |
| Total retinal thickness (µm) | 157.7±99.4 | 152.0±60.7 | 0.34 | 0.73 |

## Discussion

In this experimental study on rabbits, eyes receiving intravitreal panitumumab injections combined with ranibizumab injections did not differ from the contralateral eyes receiving intravitreal injections of Ringer´s solution with respect to histo-morphometric examinations of the various retinal layers and the retina as a whole, with respect to IOP, and with respect to results of intravital examinations of the anterior chamber and fundus.

The results of our study cannot directly be compared with other studies since the intraocular tolerability and safety of a combined application of panitumumab as an EGF receptor blocker and ranibizumab as an VEGF antibody have not been

**Table 2. Intraocular pressure in rabbits before and shortly after intravitreal injections of panitumumab combined with ranibizumab (right eyes) and of only ranibizumab (left eyes).**

| Intraocular pressure (IOP) (mm Hg) | Right eyes (study group) | Left eyes (control group) | P-value |
|---|---|---|---|
| Baseline | 7.3 ± 1.4 | 7.9 ± 0.8 | 0.28 |
| After first injection | 30.0 ± 17.3 | 28.7 ± 16.2 | 0.50 |
| Before second injection | 7.5 ± 0.8 | 8.1 ± 1.6 | 0.31 |
| After second injection | 37.2 ± 16.9 | 28.3 ± 16.4 | 0.13 |
| Before third injection | 7.6 ± 0.9 | 7.6 ± 0.7 | 1.00 |
| After third injection | 29.8 ± 16.6 | 33.1 ± 16.2 | 0.67 |
| IOP difference after injection minus before injection | | | |
| First injection | 23.0 ± 16.8 | 15.0 ± 16.1 | 0.41 |
| Second injection | 29.7 ± 17.0 | 20.1 ± 16.3 | 0.12 |
| Third injection | 22.2 ± 16.1 | 25.4 ± 15.9 | 0.93 |

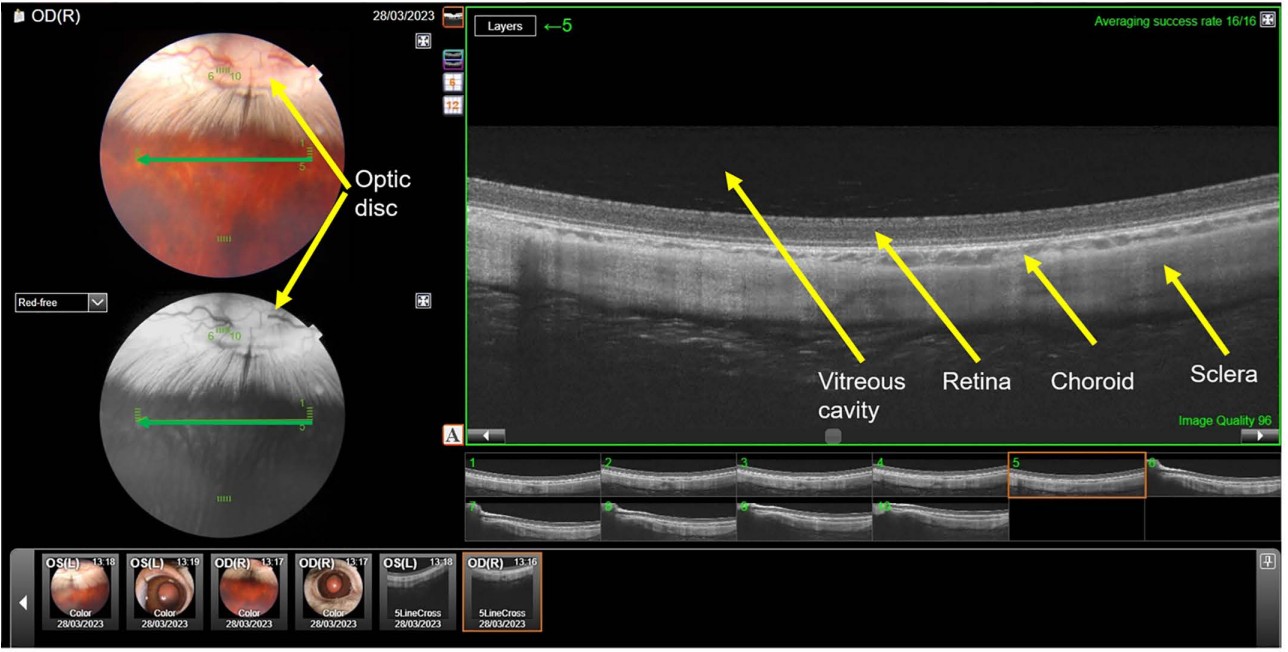

**Fig 4. Color fundus photograph (upper left part of the presentation), infrared fundus image (lower left image of the presentation), and optical coherence tomographic (OCT) images of the retina (right side of the presentation, with one OCT in higher magnification in the upper right part) of a rabbit after repeated intravitreal combined injections of panitumumab (1 mg in 0.10 mL) ranibizumab (0.5 mg in 0.05 mL) in intervals of 4 weeks.** Anterior segment photographs in low magnification in the lower part of the presentation. The green line in the clinical fundus photograph and the infrared image shows the orientation of the OCT section shown in the upper right part of the presentation.

examined yet. The findings of our study agree with the observations made in recent experimental and clinical studies with intravitreally injected panitumumab in patients and with intravitreally injected cetuximab as another EGF receptor blocker in rabbits.[21,22] In these studies, a dose of.5 mg cetuximab (0.10 mL; 5 mg cetuximab/mL) was three times injected in four-weekly intervals into the eyes of young and old rabbits, and panitumumab in doses of 0.6 mg, 1.2 mg, and 1.8 mg was repeatedly injected into the eyes of adult highly myopic patients with myopic macular degeneration [21,22]. In

both studies, the repeated injections were well tolerated. In the case of the rabbits, the injected eyes as compared to the contralateral eyes with intravitreal applications of Ringer´s solution did not differ significantly in ocular biometric measurements of axial length and anterior chamber depth, IOP, signs of intraocular inflammation or fundus peculiarities, and thickness of retinal layers and the total retina as measured histomorphometrically in various regions of the globe [21]. In the clinical study, none of the patients developed treatment-related systemic adverse events or signs of intraocular inflammatory reactions, with no significant differences in best corrected visual acuity and IOP as measured at baseline and at the end of the study [22]. In the clinical study as a phase-1 trial, the adult highly myopic patients had received the intravitreal panitumumab injections based on the hypothesis, that adult highly myopic eyes can undergo further axial elongation, that such an axial elongation is a major risk factor for deterioration of myopic macular degeneration, and that the myopic axial elongation is related to a retina-triggered, EGF-induced and RPE-mediated growth of Bruch´s membrane in the midperiphery of the fundus [18,23]. The findings of our study also agree with numerous investigations on the safety (and effect) of repeated intraocular applications of ranibizumab for the therapy of intraocular neovascular disorders such as exudative age-related macular degeneration and proliferative diabetic retinopathy [5–7].

Our study adds information to the current knowledge by examining the intraocular side-effects of a combined application of panitumumab and ranibizumab. The increased volume of combined 0.15 mL of the intravitreal injections was associated with an increase in IOP by approximately 20 mm Hg to 30 mm Hg without a statistically significant inter-eye difference in the rabbits. The larger combined volume of two different antibodies injected intraocularly in the present study as compared to the previous investigations which were focused only on either ranibizumab or panitumumab was not associated with a detected increased incidence of signs of intraocular inflammation or retinal tissue affection, such as a decrease in the cell count of the retinal ganglion cells (Table 1).

The potential clinical importance of the findings obtained in the present study is the combined intravitreal application of ranibizumab together with panitumumab in patients with exudative AMD, polypoidal choroidal vasculopathy and neovascular myopic macular degeneration. All these three entities belong to the group of exudative maculopathies in which newly formed blood vessels grow into the subretinal space and initiate the formation of a subretinal fibrotic or fibrosis-like scar. Up to now, only the neovascular component of these neovascular maculopathies has been addressed (by the application of anti-VEGF drugs), while the fibrotic component has been mostly neglected so far. Histological studies have suggested that the subretinal tissue is formed by proliferating RPE cells [9]. Cell culture studies revealed that migration and proliferation of RPE cells in vitro could be increased by growth factors like EGF and reduced by application of corresponding antibodies, like EGF receptor blockers [10–18]. Other investigations reported about the presence of cellular receptors for EGF and other growth factors on the RPE cell surface [10–18]. Also, abnormally high intraocular concentration and amount of EGF and EGF family members were found in eyes with exudative AMD and highly myopic eyes with myopic macular neovascularization [19,20]. Fitting with these results, a laser-induced RPE lesion in an experimental study on rabbits led to a smaller hyperpigmented scar if panitumumab as EGF receptor blocker had intravitreally been injected [24]. Combining the observations made in these various investigations may make one infer that an intravitreally applied EGF receptor blocker like panitumumab combined with ranibizumab could be helpful to reduce the formation of subretinal scars secondary to the subretinal neovascularization.

Limitations of the present investigation have to be discussed. First, as for any experimental animal study, species differences between humans and rabbits have to be taken into account. Panitumumab is a fully human monoclonal IgG2-antibody which is produced by recombinant DNA-technology in a mammal cell line [25]. Its activity to block the EGF-receptor in rabbits will be lower than in humans, so that if applied in patients, the effect of panitumumab may be higher than in the present study. Second, species differences between rabbits and men in the anatomy and physiology of the macula prevent a direct transfer of the findings made in the rabbits of our study onto the clinical situation. Third, the study duration was relatively short, so that observations on the long-term safety were not made. Fourth, it has been known that intravitreally applied ranibizumab is generally well tolerated by the eye, so that the eyes of the control group should have

received ranibizumab combined with Ringer's solution. The advantage of applying only Ringer's solution into the eyes of the control group was that an overall potentially toxic effect of the combined ranibizumab/ panitumumab application could be assessed and compared against a purely negative control group. Fifth, repeated tonometric measurements over time as compared to a single IOP reading would have provided a better safety assessment with respect to an IOP change induced by panitumumab. Sixth, the study sample size was small, so that a difference between study group and control group would have had to be large to reach statistical significance. In view of this major limitation of the present study, one may only be able to state that the observed lack of a statistically significant difference between study group and control group with respect to IOP and histomorphometry suggests the absence of a major tissue damage in the study group as compared to the control group.

## Conclusions

In conclusion, repeated intraocular combined applications of panitumumab (1.0 mg) and ranibizumab (0.05 mg) were well tolerated in rabbits and may open the possibility to discuss future clinical trials exploring in patients the intraocular safety of a combined application of panitumumab and ranibizumab and eventually the potential efficacy of such a combined application for the prevention of subretinal RPE proliferations in eyes with neovascular macular disorders such as exudative age-related macular degeneration.

## Author contributions

**Conceptualization:** Mukharram M. Bikbov, Gyulli M. Kazakbaeva, Songhomitra Panda-Jonas, Jost B. Jonas.

**Data curation:** Mukharram M. Bikbov, Gyulli M. Kazakbaeva, Iskander D. Valishin, Aigul M. Ishbulatova, Songhomitra Panda-Jonas, Gulshat R. Mustafina, Jost B. Jonas.

**Formal analysis:** Gyulli M. Kazakbaeva, Iskander D. Valishin, Aigul M. Ishbulatova, Gulshat R. Mustafina, Jost B. Jonas.

**Funding acquisition:** Mukharram M. Bikbov.

**Investigation:** Mukharram M. Bikbov, Gyulli M. Kazakbaeva, Iskander D. Valishin, Aigul M. Ishbulatova, Songhomitra Panda-Jonas, Gulshat R. Mustafina, Jost B. Jonas.

**Methodology:** Mukharram M. Bikbov, Gyulli M. Kazakbaeva, Aigul M. Ishbulatova, Songhomitra Panda-Jonas, Gulshat R. Mustafina, Jost B. Jonas.

**Project administration:** Mukharram M. Bikbov, Gyulli M. Kazakbaeva.

**Supervision:** Mukharram M. Bikbov, Gyulli M. Kazakbaeva, Jost B. Jonas.

**Validation:** Mukharram M. Bikbov, Gyulli M. Kazakbaeva, Iskander D. Valishin, Aigul M. Ishbulatova, Songhomitra Panda-Jonas, Gulshat R. Mustafina, Jost B. Jonas.

**Visualization:** Mukharram M. Bikbov, Gyulli M. Kazakbaeva, Iskander D. Valishin, Aigul M. Ishbulatova, Songhomitra Panda-Jonas, Gulshat R. Mustafina, Jost B. Jonas.

**Writing – original draft:** Songhomitra Panda-Jonas, Jost B. Jonas.

**Writing – review & editing:** Mukharram M. Bikbov, Gyulli M. Kazakbaeva, Iskander D. Valishin, Aigul M. Ishbulatova, Songhomitra Panda-Jonas, Gulshat R. Mustafina, Jost B. Jonas.

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
