## [Decision Letter · Decision Letter 0]

1 Sep 2025

Dear Dr. Jonas,

Thank you for submitting your manuscript to PLOS ONE. After careful consideration, we feel that it has merit but does not fully meet PLOS ONE’s publication criteria as it currently stands. Therefore, we invite you to submit a revised version of the manuscript that addresses the points raised during the review process.

We look forward to receiving your revised manuscript.

Kind regards,

Yalong Dang

Academic Editor

PLOS ONE

Journal Requirements:

2. To comply with PLOS One submissions requirements, in your Methods section, please provide additional information regarding the experiments involving animals and ensure you have included details on (1) methods of sacrifice (including setup and details regarding deep anaesthesia, (2) efforts to alleviate suffering.

4. In the online submission form, you indicated that the datasets used and/or analyzed during the current study available from the corresponding author on reasonable request.

Jost B. Jonas, Songhomitra Panda-Jonas: European patent EP 3 271 392, JP 2021-119187, and US 2021 0340237 A1: “Agents for use in the therapeutic or prophylactic treatment of myopia or hyperopia; European patent application 23196899.1 “EGFR Antagonists for the treatment of diseases involving unwanted migration, proliferation, and metaplasia of retinal pigment epithelium (RPE) cells“. All other authors: None

Reviewers' comments:

Reviewer's Responses to Questions

**Comments to the Author**

1. Is the manuscript technically sound, and do the data support the conclusions?

Reviewer #1: Partly

Reviewer #2: Partly

Reviewer #3: No

2. Has the statistical analysis been performed appropriately and rigorously?

Reviewer #1: Yes

Reviewer #2: Yes

Reviewer #3: I Don't Know

3. Have the authors made all data underlying the findings in their manuscript fully available?

Reviewer #1: Yes

Reviewer #2: No

Reviewer #3: No

4. Is the manuscript presented in an intelligible fashion and written in standard English?

Reviewer #1: Yes

Reviewer #2: Yes

Reviewer #3: Yes

Reviewer #1: The sample size of 8 rabbits (4 per time point) may be limiting for statistical analysis. A larger sample size and blinding could improve the study's robustness. Using Ringer's solution instead of ranibizumab as the control makes it challenging to attribute effects specifically to panitumumab. The Wilcoxon test with 4 animals per group may have limited power to detect differences.

Suggestions for Improvement:

IOP monitoring: Repeated measurements over time would provide better safety assessment.

Statistical analysis: The adding of analysis of variance components may be favorable.

Clinical discussion: Expanding on potential clinical implications would strengthen the manuscript.

Reviewer #2: Dear Editor,

This manuscript is a very simple study on the potential toxicity of multiple antibody injections in a rabbit model. Overall, the manuscript is well written, and needs only minor copy-editing for grammar. E.g., line 177 “injections were performed in anesthesia” should be corrected to “injections were performed under anesthesia”. Line 169: “Examinations of the anterior chamber neither revealed signs of intraocular inflammation” needs minor correction for clarity. Statistical analysis is appropriate.

Although straightforward, and generally well designed, this study is missing data would greatly strengthen the authors case that these treatments are safe. This is particularly important since the n for each experimental time point is quite low (n=4, 8 rabbits total). Namely:

- Including representative images for the control eyes, a positive control, and a negative control for the TUNEL labeling. Include labels of the retinal layers on the figures.

- Including representative images of the H&E stained retinas – both the control and the antibody-injected eyes for comparison. Include labels for the retinal layers.

- Including representative images for the control eyes for the fundus and OCT data. Labeling the retina layers.

- Including imaging data for the anterior segment of the eye (line 169).

As for data availability, the authors have opted to not share the data – i.e., they selected No, some restrictions will apply. I don’t understand why, as it’s a very simple study and it should be quite easy to make all of the data, for only 8 rabbits, available. In fact, sharing this data would strengthen the merits and claims of this study.

I think that this study should not be published until more data are shown and the figures reflecting these data amended. I chose “minor revisions” because these data should exist, and I am not requesting more experiments, just showing more data from the experiments that were done. A simple additional experiment that could be performed would be to label eyes for microglial infiltration (Iba1), which would potentially signal early biological signs of inflammation.

Reviewer #3: The manuscript examines the potential effect of combined intraocular injection of panitumumab and ranibizumab on the rabbit retina. Such a “safety” study is valuable given the potential benefit of adding an EGF receptor antibody (panitumumab) to the well-established effect of VEGF antibodies (e.g. ranibizumab) as a treatment for exudative maculopathies. Whereas VEGF antibodies reduce neovascularization, they do not reverse the formation of an irreversible subretinal fibrous scar due to RPE proliferation. In vitro studies have suggested that EGF promotes RPE proliferation, an effect that can be blocked by EGF receptor blockers. Clinical studies have suggested that intraocular EGF levels are elevated in exudative age-related macular degeneration and myopic macular neovascularization. An obvious idea would be to employ a combined treatment, with blockers of VEGF and EGF receptors used concurrently. This current manuscript suggests that an important first step is to test the safety of such a combined treatment using an animal model (rabbit).

The study employs four measures to compare rabbit eyes injected with either the combination of panitumumab and ranibizumab (right eye) or saline (left eye): 1) histological measures of retinal layer composition (cell counts) and thickness; 2) intraocular pressure (IOP); 3) TUNEL-labelling to detect apoptosis; and 4) fundus photography and OCT.

Taken as a whole, the results suggest that the combination of panitumumab and ranibizumab does not produce any signs of retinal damage: there were no statistically significant differences between the treated (right) and control (left) eye with respect to retinal layer thickness, mcell number or IOP; there was no evidence of TUNEL-labelling in treated or control retina; there were no signs of changes visualized using fundus photography and OCT.

The results, as presented, have several weaknesses that limit the suitability of the manuscript:

1. No photomicrographs of the retinal sections are provided to allow the reader to evaluate the quality of the histology.

2. Given the large variation inherent in the histological measurements and the sample size, it is not clear if the sensitivity is sufficient. At the very least, some discussion about what would constitute a meaningful change in the parameters would be informative.

3. In Table 1, two P-values of 0.04 are reported. Why were these not considered significant, given that alpha was set at 0.05?

4. The TUNEL data is weak without a positive control, demonstrating that the method employed by the authors is capable of detecting TUNEL+ cells.

5. Figure 2 is inadequate. What features in these images were considered important, and there should be examples comparing these parameters in treated vs. untreated eyes. The description of the results related to the figure is inadequate. What signs of “intraretinal inflammation”? What “other changes”? Why not use OCT to perform retina thickness measurements that would add to the histology?

**Do you want your identity to be public for this peer review?** For information about this choice, including consent withdrawal, please see our Privacy Policy

Reviewer #1: No

Reviewer #2: No

Reviewer #3: No

---

## [Author Response · Author response to Decision Letter 1]

2 Nov 2025

Changes Made in Response to the Reviewers´ Comments

(Re.: PONE-D-25-38703: Safety of Intravitreal Panitumumab Combined with Intravitreal Ranibizumab in Rabbits)

Journal Requirements:

1. Journal Requirements: When submitting your revision, we need you to address these additional requirements.

Our response: The revised manuscript has been formatted according to the Instructions for Authors

2. Journal Requirements: 2. To comply with PLOS One submissions requirements, in your Methods section, please provide additional information regarding the experiments involving animals and ensure you have included details on (1) methods of sacrifice (including setup and details regarding deep anaesthesia, (2) efforts to alleviate suffering.

Our response: It has been stated in the revised manuscript: “At the time of sacrificing the animals, the eyes were enucleated under deep general anesthesia, and the rabbits were sacrificed by injecting air (5–10 mL) into the ear vein.” (Page 4, line 131)

3. Journal Requirements: 3. We note that you have indicated that there are restrictions to data sharing for this study. PLOS only allows data to be available upon request if there are legal or ethical restrictions on sharing data publicly. For more information on unacceptable data access restrictions, please see http://journals.plos.org/plosone/s/data-availability#loc-unacceptable-data-access-restrictions.

Our response: We have uploaded the datafile as a supporting material file.

4. Journal Requirements: 4. In the online submission form, you indicated that the datasets used and/or analyzed during the current study available from the corresponding author on reasonable request.

Our response:

- We have uploaded the datafile as a supporting material file.

- It has been stated in the revised manuscript: “Data Availability Statement: The dataset including all data necessary to repeat the statistical analysis has been provided together with this manuscript” (Page 8, line 870)

5. Journal Requirements: 5. Thank you for stating the following in the Competing Interests section:

Jost B. Jonas, Songhomitra Panda-Jonas: European patent EP 3 271 392, JP 2021-119187, and US 2021 0340237 A1: „Agents for use in the therapeutic or prophylactic treatment of myopia or hyperopia; European patent application 23196899.1 „EGFR Antagonists for the treatment of diseases involving unwanted migration, proliferation, and metaplasia of retinal pigment epithelium (RPE) cells“. All other authors: None

Our response: It has been written in the cover letter (and on the title page of the revised manuscript): “It is also declared: Conflict of interest disclosure: Jost B. Jonas, Songhomitra Panda-Jonas: European patent EP 3 271 392, JP 2021-119187, and US 2021 0340237 A1: „Agents for use in the therapeutic or prophylactic treatment of myopia or hyperopia; European patent application 23196899.1 „EGFR Antagonists for the treatment of diseases involving unwanted migration, proliferation, and metaplasia of retinal pigment epithelium (RPE) cells“. This does not alter our adherence to PLOS ONE policies on sharing data and materials. All other authors: None.” (Page 1, line 30)

6. Journal Requirements: 6. If the reviewer comments include a recommendation to cite specific previously published works, please review and evaluate these publications to determine whether they are relevant and should be cited. There is no requirement to cite these works unless the editor has indicated otherwise.

Our response: None of the reviewers recommended citation of a specific previous article.

Reviewers' comments:

Reviewer's Responses to Questions

Comments to the Author

1. Is the manuscript technically sound, and do the data support the conclusions? The manuscript must describe a technically sound piece of scientific research with data that supports the conclusions. Experiments must have been conducted rigorously, with appropriate controls, replication, and sample sizes. The conclusions must be drawn appropriately based on the data presented.

Reviewer #1: Partly

Reviewer #2: Partly

Reviewer #3: No

2. Has the statistical analysis been performed appropriately and rigorously?

Reviewer #1: Yes

Reviewer #2: Yes

Reviewer #3: I Don't Know

3. Have the authors made all data underlying the findings in their manuscript fully available?

Reviewer #1: Yes

Reviewer #2: No

Reviewer #3: No

4. Is the manuscript presented in an intelligible fashion and written in standard English?

Reviewer #1: Yes

Reviewer #2: Yes

Reviewer #3: Yes

5. Review Comments to the Author; Please use the space provided to explain your answers to the questions above. You may also include additional comments for the author, including concerns about dual publication, research ethics, or publication ethics. (Please upload your review as an attachment if it exceeds 20,000 characters)

7. Reviewer #1: The sample size of 8 rabbits (4 per time point) may be limiting for statistical analysis. A larger sample size and blinding could improve the study's robustness. Using Ringer's solution instead of ranibizumab as the control makes it challenging to attribute effects specifically to panitumumab. The Wilcoxon test with 4 animals per group may have limited power to detect differences.

Suggestions for Improvement:

IOP monitoring: Repeated measurements over time would provide better safety assessment.

Our response: It has been stated in the revised Discussion: “Limitations of the present investigation have to be discussed. First, … . Fifth, repeated tonometric measurements over time as compared to a single IOP reading would have provided a better safety assessment with respect to an IOP change induced by injections.” (Page 7, line 253)

8. Reviewer #1: Statistical analysis: The adding of analysis of variance components may be favorable.

- Our response: For the revision of the manuscript, we additionally conducted an ANOVA: “In the statistical analysis (SPSS for Windows, version 27.0, IBM-SPSS, Chicago, IL, USA), we assessed the mean values ± their standard deviations of the main outcome parameters, i.e., IOP, and histomorphometric parameters. Using the Wilcoxon test for paired samples, we compared the measurements obtained in the right eyes versus those of the left eyes. An analysis of variance was additionally performed. A two-sided P-value was considered statistically significant if it was smaller than 0.05.” (Page 4, line 145)

- “In an analysis of variance, similar findings were made (Table 1).” (Page 5, line 162)

9. Reviewer #1: Clinical discussion: Expanding on potential clinical implications would strengthen the manuscript.

Our response: It has been added to the revised Discussion: “The potential clinical importance of the findings obtained in the present study is the combined intravitreal application of ranibizumab together with panitumumab in patients with exudative AMD, polypoidal choroidal vasculopathy and neovascular myopic macular degeneration. All these three entities belong to the group of exudative maculopathies in which newly formed blood vessels grow into the subretinal space and initiate the formation of a subretinal fibrotic or fibrosis-like scar. Up to now, only the neovascular component of these neovascular maculopathies has been addressed (by the application of anti-VEGF drugs), while the fibrotic component has been mostly neglected so far. Histological studies have suggested that the subretinal tissue is formed by proliferating RPE cells [9]. Cell culture studies revealed that migration and proliferation of RPE cells in vitro could be increased by growth factors like EGF and reduced by application of corresponding antibodies, like EGF receptor blockers [10-18]. Other investigations reported about the presence of cellular receptors for EGF and other growth factors on the RPE cell surface [10-18]. Also, abnormally high intraocular concentration and amount of EGF and EGF family members were found in eyes with exudative AMD and highly myopic eyes with myopic macular neovascularization [19,20]. Fitting with these results, a laser-induced RPE lesion in an experimental study on rabbits led to a smaller hyperpigmented scar if panitumumab as EGF receptor blocker had intravitreally been injected [24]. Combining the observations made in these various investigations may make one infer that an intravitreally applied EGF receptor blocker like panitumumab combined with ranibizumab could be helpful to reduce the formation of subretinal scars secondary to the subretinal neovascularization.” (Page 6, line 222)

Reviewer #2:

10. Reviewer #2: Dear Editor, This manuscript is a very simple study on the potential toxicity of multiple antibody injections in a rabbit model. Overall, the manuscript is well written, and needs only minor copy-editing for grammar. E.g., line 177 “injections were performed in anesthesia” should be corrected to “injections were performed under anesthesia”. Line 169: “Examinations of the anterior chamber neither revealed signs of intraocular inflammation” needs minor correction for clarity. Statistical analysis is appropriate.

Our response: The sentences have been re-worded:

- “The injections were performed under general anesthesia .. .” (Page 4, line 118)

- “Biomicroscopical examinations of the anterior chamber neither revealed any signs of intraocular inflammation.“ (Page 5, line 179)

11. Reviewer #2: Although straightforward, and generally well designed, this study is missing data would greatly strengthen the authors case that these treatments are safe. This is particularly important since the n for each experimental time point is quite low (n=4, 8 rabbits total). Namely:

- Including representative images for the control eyes, a positive control, and a negative control for the TUNEL labeling. Include labels of the retinal layers on the figures.

- Including representative images of the H&E stained retinas – both the control and the antibody-injected eyes for comparison. Include labels for the retinal layers.

- Including representative images for the control eyes for the fundus and OCT data. Labeling the retina layers.

- Including imaging data for the anterior segment of the eye (line 169).

Our response:

- In the revised manuscript, we have added more examples of TUNEL-stained images and a positive TUNEL control image. (Pages 17-20)

- An anterior ocular segment photograph is shown now on Fig. 4.

-In the revised manuscript we have added H&E-stained histological photographs of the retina taken from different eyes and ocular regions. (Pages 13-16)

- A representative image of the anterior segment and fundus including color fundus photography, infrared image and optical coherence tomographic images of the retina has been added and has been labelled showing the different tissue layers (Fig. 4) (Page 21).

12. Reviewer #2: As for data availability, the authors have opted to not share the data – i.e., they selected No, some restrictions will apply. I don’t understand why, as it’s a very simple study and it should be quite easy to make all of the data, for only 8 rabbits, available. In fact, sharing this data would strengthen the merits and claims of this study.

Our response: As also pointed out in the reply to comment # 3 by “Journal Requirements”, it has been stated now: “Data Availability Statement: The dataset including all data necessary to repeat the statistical analysis has been provided together with this manuscript” (Page 8, line 870)

13. Reviewer #2: I think that this study should not be published until more data are shown and the figures reflecting these data amended. I chose “minor revisions” because these data should exist, and I am not requesting more experiments, just showing more data from the experiments that were done. A simple additional experiment that could be performed would be to label eyes for microglial infiltration (Iba1), which would potentially signal early biological signs of inflammation.

Our response: As described above in detail, additional data have been provided by submitting the datafile with the microdata, and new figures and images have been added to the revised manuscript. While the authors fully agree with the reviewer about the potential value of an additional assessment of activated microglial cells, the authors unfortunately cannot follow the recommendation, since for the time being they do not have access to

---

## [Decision Letter · Decision Letter 1]

27 Nov 2025

Safety of íntravitreal panitumumab combined with intravitreal ranibizumab in rabbits

PONE-D-25-38703R1

Dear Dr. Jonas,

We’re pleased to inform you that your manuscript has been judged scientifically suitable for publication and will be formally accepted for publication once it meets all outstanding technical requirements.

Kind regards,

Yalong Dang

Academic Editor

PLOS ONE

Additional Editor Comments (optional):

The authors well addresseed the concerns raised by the three reviewer, I think it would be acceptable in its current form for publication.

Reviewers' comments:

Reviewer's Responses to Questions

**Comments to the Author**

Reviewer #1: All comments have been addressed

2. Is the manuscript technically sound, and do the data support the conclusions?

Reviewer #1: Yes

3. Has the statistical analysis been performed appropriately and rigorously?

Reviewer #1: Yes

4. Have the authors made all data underlying the findings in their manuscript fully available?

Reviewer #1: Yes

5. Is the manuscript presented in an intelligible fashion and written in standard English?

Reviewer #1: Yes

Reviewer #1: The manuscript is now in strong shape. The reasoning is clear, and the study presents meaningful scientific implications. The revised statistical analyses are more robust, especially with the inclusion of the new figures—such as the TUNEL-stained images, the positive TUNEL control, and the anterior ocular segment photograph—which enhance the reliability and clarity of the results. The added statistical details further strengthen the overall rigor of the work.

The authors have also appropriately discussed the study’s limitations and outlined directions for future research, making the manuscript as comprehensive as the experimental conditions reasonably allow while paving the way for relevant follow-up investigations. As all datasets now align with PLOS ONE’s requirements, the paper is suitable for publication.

**Do you want your identity to be public for this peer review?** For information about this choice, including consent withdrawal, please see our Privacy Policy

Reviewer #1: No

---

## [Editor Report · Acceptance letter]

PONE-D-25-38703R1

PLOS One

Dear Dr. Jonas,

I'm pleased to inform you that your manuscript has been deemed suitable for publication in PLOS One. Congratulations! Your manuscript is now being handed over to our production team.

Kind regards,

on behalf of

Dr Yalong Dang

Academic Editor

PLOS One